# Comparative Mitogenomics of Flesh Flies: Implications for Phylogeny

**DOI:** 10.3390/insects13080718

**Published:** 2022-08-10

**Authors:** Jin Shang, Wentian Xu, Xiaofang Huang, Dong Zhang, Liping Yan, Thomas Pape

**Affiliations:** 1School of Ecology and Nature Conservation, Beijing Forestry University, Beijing 100083, China; 2Natural History Museum of Denmark, University of Copenhagen, Universitetsparken 15, DK-2100 Copenhagen, Denmark

**Keywords:** mitogenome, sarcophagidae, phylogeny, evolution

## Abstract

**Simple Summary:**

Flesh flies (Diptera: Sarcophagidae) make up the second-largest family of Oestroidea. They are well known for their veterinary, forensic, and medical importance due to their extremely diverse feeding habits, making them a hotspot in dipterology research. Therefore, efforts have been devoted to accumulating the mitochondrial genomes of Sarcophagidae. However, the mitogenome of flesh flies has mostly been sequenced for the genus *Sarcophaga* of the subfamily Sarcophaginae, and the other two subfamilies, Miltogramminae and Paramacronychiinae, have barely been touched. Here, we sequenced the mitochondrial genomes of five sarcophagid species from four genera representing all three subfamilies and investigated the phylogeny and evolution of flesh flies from the perspective of the mitogenome via comprehensive comparative analyses employing all mitogenomic data. This study will broaden our knowledge of flesh flies and make contributions to forensics, veterinary science, entomology, and ecology.

**Abstract:**

Flesh flies (Diptera: Sarcophagidae) represent a rapid radiation belonging to the Calyptratae. With more than 3000 known species, they are extraordinarily diverse in terms of their breeding habits and are therefore of particular importance in human and veterinary medicine, forensics, and ecology. To better comprehend the phylogenetic relationships and evolutionary characteristics of the Sarcophagidae, we sequenced the complete mitochondrial genomes of five species of flesh flies and performed mitogenomic comparisons amongst the three subfamilies. The mitochondrial genomes match the hypothetical condition of the insect ancestor in terms of gene content and gene arrangement. The evolutionary rates of the subfamilies of Sarcophagidae differ significantly, with Miltogramminae exhibiting a higher rate than the other two subfamilies. The monophyly of the Sarcophagidae and each subfamily is strongly supported by phylogenetic analysis, with the subfamily-level relationship inferred as (Sarcophaginae, (Miltogramminae, Paramacronychiinae)). This study suggests that phylogenetic analysis based on mitochondrial genomes may not be appropriate for rapidly evolving groups such as Miltogramminae and that the third-codon positions could play a considerable role in reconstructing the phylogeny of Sarcophagidae. The protein-coding genes ND2 and ND6 have the potential to be employed as DNA markers for species identification and delimitation in flesh flies.

## 1. Introduction

Sarcophagidae, or flesh flies, which comprise more than 3000 known species, represent one of the most species-rich families of Calyptratae [1], and they have recently been the focus of several comprehensive studies [2,3,4,5,6,7]. They are widely distributed on all continents except Antarctica [2,4] and are divided into three subfamilies, namely, Miltogramminae, Paramacronychiinae, and Sarcophaginae [4]. Flesh flies are of great ecological and economic importance because of their diverse feeding habits. Necrophagous species breeding in animal corpses and excreta play an essential role in forensic cases [8,9,10], and agents of myiasis have considerable importance for human and veterinary medicine [11,12]. Lower Miltogramminae are sarcosaprophagous while the higher Miltogramminae are kleptoparasites of Hymenoptera, with larvae developing in the nests of their hosts, feeding on the food stored for the wasp or bee progeny [6,13,14]. Paramacronychiinae are mostly parasitoids or predators of invertebrates, but the genera *Sarcophila* and *Wohlfahrtia* contain scavengers and agents of myiasis [15]. For example, *Agria* contains parasitoids and predators of insects [16,17], and some species of *Wohlfahrtia* produce traumatic myiasis in humans and livestock [18,19]. Sarcophaginae are the largest subfamily within Sarcophagidae. They are biologically diverse flies and are parasitoids or predators of invertebrates and scavengers of carrion and dung or feces [2,4].

The mitochondrial genome of insects is a small-sized organelle genome that is inherited maternally and seldom recombined [20]. Due to its small size, large number of copies, conserved genome components, fast evolutionary rate, and uncommon recombination, the mitochondrial genome has become a formidable tool in phylogenetic and evolutionary studies of various insect groups [20,21,22,23,24,25,26,27,28]. In comparative studies, mitogenome features such as compositional bias and substitutional rate variation provide essential information for the reconstruction of phylogeny as a basis for tracking evolution among insects [24,26,29,30,31,32]. Despite the fast accumulation of sequenced mitogenomes generated by high-throughput sequencing [33], our understanding of mitogenomic evolution in insects remains fragmentary.

To improve our understanding of the diversity in the mitochondrial genome and the evolution of mitochondrial genetics of Sarcophagidae, we sequenced the mitochondrial genomes of five sarcophagid species belonging to four genera and representing all three subfamilies. The genomic structure, base composition, AT content, substitutional and evolutionary rates, and the compositional heterogeneity of the mitogenomes were investigated, and the phylogeny of Sarcophagidae was reconstructed based on all mitogenomes available for flesh flies from GenBank.

## 2. Materials and Methods

### 2.1. Sampling and Taxa Identification

Specimens of *Taxigramma karakulensis* (Miltogramminae), *Wohlfahrtia balassogloi* and *Wohlfahrtia fedtschenkoi* (Paramacronychiinae), and *Blaesoxipha lapidosa* (Sarcophaginae) from the Kalamaili Nature Reserve in Xinjiang and *A. mihalyii* (Paramacronychiinae) from Songshan in Beijing were collected with a hand net and identified following Ye [34] and Xue et al. [35] (Appendix A). The specimens were stored at −20 °C in 96% ethanol until DNA was extracted. Classification followed the most recent world catalogue [4] with subsequent updates [2,3,7].

### 2.2. DNA Extraction, Sequencing and Assembly

A Tiangen DNeasy Blood and Tissue Kit (Tiangen, Beijing, China) was used to extract genomic DNA from muscle tissue of each specimen. Among them, *Blaesoxipha lapidosa* and *Taxigramma karakulensis* were sequenced in combination, while *Agria mihalyii*, *Wohlfahrtia balassogloi*, and *Wohlfahrtia fedtschenkoi* were sequenced separately. A segment of mitochondrial gene COI was amplified and sequenced for each species by polymerase chain reaction (PCR) with universal primers as the “reference sequence” as described in Yan et al. [36]. Subsequently, the genomic DNA Library was prepared pooling the extracted genomic DNA and sequenced with a NovaSeq 6000 (PE150; Illumina, San Diego, CA, USA) platform. Trimmomatic [37] was used to trim raw data as described in Yan et al. [2] before assembly using IDBA_UD v 1.1.1 [38]. The assembly proceeded with similarity set to be 98%. The mitochondrial genomes were then extracted by the BLAST program [39] with the COI sequences as the “bait” [23].

### 2.3. Mitogenome Annotation and Sequence Analyses

Annotation of protein-coding genes (PCGs) and ribosomal RNA (rRNA) genes was performed using Geneious by comparing to homologous genes in other Calyptratae flies. The MITOS online (http://mitos2.bioinf.uni-leipzig.de/index.py, accessed on 5 May 2022) server and the tRNAscan-SE (http://lowelab.ucsc.edu/tRNAscan-SE/, accessed on 5 May 2022) online search server were used to determine the transfer RNA (tRNA) genes. A circular map of the mitochondrial genome was generated using the online server CGView (https://proksee.ca/, accessed on 7 May 2022).

All sarcophagid mitochondrial genomic data from GenBank were collected and used for comparative mitogenomic analyses together with the five newly sequenced mitogenomes in the present study (Table 1). We used MEGA X [40] to examine the base composition and codon usage of PCGs. To assess mitochondrial strand asymmetry of 13 protein-coding genes in all sarcophagid mitochondrial genomes, the AT skew and the GC skew were calculated with the following formulae: AT skew = (A − T)/(A + T) and GC skew = (G − C)/(G + C).

The nonsynonymous substitutions (Ka) and synonymous substitutions (Ks) of the PCGs were calculated using DnaSP v6.12.03 (Barcelona, Spain) [41]. The genetic distances were calculated using MEGA X with a Kimura 2-parameter model. The Xia model test in DAMBE software [42] was used to evaluate the substitution saturation of each codon position. Bowker’s test of symmetry was performed with SymTest v2.0.47 (San Francisco, CA, USA) [43] to investigate the heterogeneous sequence divergence within the dataset, and heatmaps were generated from the calculated *p*-values.

### 2.4. Phylogenetic Analysis

The phylogeny reconstruction was conducted using all available sarcophagid mitogenomes (Table 1). *Fannia scalaris* (Fanniidae), *Musca domestica* (Muscidae), *Calliphora vomitoria* (Calliphoridae), and *Delia antiqua* (Anthomyiidae) were included as outgroups, and *F. scalaris* was selected to root the tree.

Multiple sequence alignments (MSAs) of all of the protein-coding and rRNA genes were obtained by aligning each gene using the online version of MAFFT v7.453 (Osaka, Japan) [44]. Four matrices were then concatenated using SequenceMatrix v1.8.1 (Science Drive, Singapore) [45]: (i) the matrix PCGsrRNA containing all 13 PCGs and 2 rRNA genes, (ii) the matrix PCGs12 containing only the 1st- and 2nd-codon positions of the 13 PCGs, (iii) the matrix PCGs containing the 13 PCGs, and (iv) the matrix PCGs12rRNA containing the 1st- and 2nd-codon positions of the 13 PCGs and 2 rRNA genes. Maximum Likelihood (ML) analyses were inferred using IQ-TREE v2.1.2 [46] on CIPRES (Cyberinfrastructure for Phylogenetic Research) Science Gateway (available at https://www.phylo.org/, accessed on 13 June 2022), with each dataset partitioned by gene. The best tree was searched using the best models evaluated by the self-implemented ModelFinder [47]. Non-parametric replicate bootstrapping with 100 repetitions was subsequently performed to evaluate the branch support. The resulting trees were visualized using the online tool iTOL [48] (available at https://itol.embl.de/upload.cgi (accessed on 15 June 2022).

## 3. Results and Discussion

### 3.1. General Features of Sarcophagid Mitogenomes

The mitochondrial genomes of five species of Sarcophagidae were successfully sequenced for the first time. The entire mitogenomes of *A. mihalyii*, *B. lapidosa*, *T. karakulensis*, *W. balassogloi*, and *W. fedtschenkoi* were 16,375 bp, 15,454 bp, 16,543 bp, 16,589 bp, and 16,643 bp in length, respectively (Figure 1). The variation in length is mainly due to the different size of the control region, as previously observed between other Sarcophagidae species [36]. Each mitogenome encoded 37 typical insect mitochondrial genes, 13 protein-coding genes (PCGs), 2 ribosomal RNA (rRNA) genes, and 22 transfer RNA (tRNA) genes. A total of 23 genes, including 9 PCGs and 14 tRNA genes, were transcribed from the majority strand (J-strand), while 14 genes were transcribed from the minority strand (N-strand). The nine PCGs transcribed from the J-strand were ATP6, ATP8, CYTB, COI, COII, COIII, ND2, ND3, and ND6, and the other four (ND1, ND4, ND4L, and ND5) were transcribed from the N-strand. Both rRNA genes (lrRNA and srRNA) were transcribed from the N-strand (Figure 1). All genes were in the same gene order and orientation as the hypothetical ancestral mitogenome, consistent with other calyptrate flies [21,22,24,49,50].

The whole mitogenome for each of *A*. *mihalyii*, *B*. *lapidosa*, *T*. *karakulensis*, *W*. *balassogloi*, and *W*. *fedtschenkoi* was biased towards A and T, with the AT content being 74.43%, 75.17%, 79.15%, 76.02% and 75.93%, respectively. Other species of Sarcophagidae also have this typical composition (Figure 2a), and all of them represent a moderate level for calyptrate flies (Appendix A). By comparing the AT content of the different PCGs for all Sarcophagidae, ND6 (82.10%) had the highest average AT content, followed by ATP8 (81.64%) and ND4L (81.33%). The average AT content of COI (67.45%) and COIII (68.08%) were the lowest.

The nucleotide skew analysis revealed negative AT skews (Figure 2b), indicating that the overall strand was asymmetric with protein-coding genes of all sarcophagid species containing less adenine (A) than thymine (T). However, GC skews were close to zero (Figure 2c), suggesting that the content of guanine (G) and cytosine (C) was not significantly different. The correlations were calculated for the AT content versus the AT skew (y = −0.0006x − 0.1109, R^2^ = 0.0125) and for the GC content versus the GC skew (y = −0.0112x + 0.2844, R^2^ = 0.6557). The results showed that, with increasing AT content, the underabundance of A relative to T increased slightly. However, with increasing GC content, the overabundance of G relative to C gradually diminished and shifted to an overabundance of C relative to G when the GC content reached a level of about 25%.

### 3.2. Protein-Coding Genes and Codon Usage

The PCGs of *A*. *mihalyii*, *B*. *lapidosa*, *T*. *karakulensis*, *W*. *balassogloi*, and *W*. *fedtschenkoi* were 11,184 bp, 11,179 bp, 11,157 bp, 11,179 bp, and 11,179 bp in length, respectively. For the five newly sequenced mitogenomes, all the PCGs started with the codon ATN (ATA/T/G). The complete termination codon TAA or TAG was utilized by most of the PCGs in these five mitogenomes, while the incomplete stop codon T was employed by COI, COII, ND4, and ND5. All three stop codons are common in the mitochondrial genomes of insects. Moreover, Ojala et al. [51] speculated that the incomplete stop codon T could be converted into the complete stop codon TAA by post-transcriptional polyadenylation.

The relative synonymous codon usage (RSCU) of each mitogenome was estimated (Appendix A). The third-codon position was more likely to be A or T than G or C. The most frequently encoded amino acids in Sarcophagidae were Ala, Arg, Gly, Leu2, Pro, Ser2, Thr, and Val, which have the greatest RSCU values, and the most commonly used codons were UUA, GCU, GGA, and GUA.

### 3.3. Ribosomal, Transfer RNA Genes, and Control Region

All 22 tRNA genes were detected and found to be discontinuously scattered throughout the mitochondrial genome in the same location as hypothesized for the ancestral insect mitogenome. The size of each tRNA gene for these newly sequenced mitogenomes was hardly different, ranging from 62 bp to 72 bp (Appendix A). The nucleotide composition of tRNA genes is biased toward A and T, with the AT content of each gene ranging from 76.04% (*B. lapidosa*) to 77.60% (*T*. *karakulensis*). The AT content of the tRNA genes was slightly higher than that of the protein-coding genes, and the combined tRNA genes exhibited positive AT skew and negative GC skew, except *B.*
*lapidosa* (Appendix A).

The two rRNA genes were located on the N-strand, with lrRNA between the tRNA^L1^ and the tRNA^V^ genes, ranging from 1322 bp (*W*. *fedtschenkoi*) to 1329 bp (*A*. *mihalyii* and *T*. *karakulensis*), and srRNA located between the tRNA^V^ gene and the control region and ranging from 783 bp (*A*. *mihalyii*) to 790 bp (*W*. *fedtschenkoi*) (Appendix A). The AT content of the lrRNA genes ranged from 80.56% (*W*. *fedtschenkoi*) to 82.77% (*T*. *karakulensis*), while that of the srRNA genes ranged from 75.89% (*W*. *balassogloi*) to 79.97% (*T*. *karakulensis*). The AT skew and GC skew of lrRNA genes for the five species were negative and positive, respectively. The srRNA genes exhibited negative AT skew for all of the newly sequenced species except *T*. *karakulensis* (0.03), and positive GC skew was observed for all five species (Appendix A).

The length of the control region of sarcophagid species varied considerably (*A. mihalyii*, *B. lapidosa*, *T. karakulensis*, *W. balassogloi*, and *W. fedtschenkoi* were 1545 bp, 650 bp, 1753 bp, 1789 bp, and 1828 bp in length, respectively), whereas there were few differences in the remaining region. Extrapolated from the information of these five species (Appendix A), the length of the mitochondrial genome of sarcophagid species mainly depends on the size of the control region, which is consistent with other insects. The AT content in the control region ranged from 83.37% (*A*. *mihalyii*) to 89.85% (*T*. *karakulensis*), which is significantly higher than for the other regions of the mitogenomes, indicating a strong AT bias in this region (Appendix A).

### 3.4. Evolutionary Rates and Heterogeneous Sequence Divergence

All three subfamilies have similar synonymous substitution rates (Ks), while the nonsynonymous substitution rates (Ka) of Miltogramminae were in all cases higher than those of Paramacronychiinae and Sarcophaginae (Figure 3). The Ka/Ks ratio (ω) is used for investigating the evolutionary rate. All 13 PCGs were suggested to be under strong purifying selection, with the ω of each gene calculated to be less than 1.0. The Ka/Ks ratio in Miltogramminae was higher than in the other two subfamilies, indicating an accelerated evolutionary rate in Miltogramminae.

Furthermore, the pairwise nonsynonymous (Ka) to synonymous (Ks) substitution ratio (ω) of all the PCGs in Sarcophagidae was analyzed, and the Ka/Ks ratio of ATP8 (ω = 0.286) and ND6 (ω = 0.203) was higher, indicating that these genes were under relatively little selection pressure (Figure 4). However, the Ka/Ks ratio of COI was the lowest (ω = 0.051), indicating a higher selection pressure for COI than for other genes. Based on the pairwise genetic distance calculation, ND6 (0.205), ND2 (0.175), and ND3 (0.174) had the highest values, while ND1 (0.105), ND4L (0.108), and ND4 (0.126) had the lowest.

The taxonomic identification of flies is still mostly limited to morphology-based identification of adults, but the identification of species with complex morphological features or extreme similarity becomes quite tricky. However, this problem can be solved at the molecular level using appropriate molecular markers and DNA barcoding techniques. Generally, the mitochondrial COI gene is the most extensively utilized molecular marker for molecular identification and phylogenetic analysis among species [52,53,54,55]. In this study, COI was the most conserved, most slowly evolving, and least variable gene among all PCGs in the Sarcophagidae. If the COI gene is proven to provide low-resolution power, then alternative genes with sufficient size, relatively high Ka/Ks ratio, and rapid evolution could be used as potential DNA markers [52,56], notably ND2 and ND6, for species identification and delimitation in flesh flies.

The heterogeneity of the sequence divergence was examined (Appendix A). The dataset PCGsrRNA showed the highest heterogeneous pairwise sequence divergence, while that of the dataset PCGs was higher than PCGs12. Therefore, the rRNA genes and third-codon positions are the main reasons for such nucleotide heterogeneity, and consequently, excluding the third-codon positions from the datasets would reduce the degree of sequence heterogeneity.

### 3.5. Phylogenetic Reconstruction

Phylogenetic analyses using the four datasets yielded similar topologies (Figure 5 and Appendix A). The monophyly of Sarcophagidae received maximum support by all reconstructions. All three subfamilies were supported as monophyletic, and their relationship was consistently inferred as (Sarcophaginae, (Paramacronychiinae + Miltogramminae)) (BSs = 95–100; Figure 5 and Appendix A). These results are in agreement with recent phylogenetic studies based on molecular data [2,3,36,49] but conflict with the older hypothesis based on morphology, which supports a sister-group relationship between Paramacronychiinae and Sarcophaginae [4,57].

The genus-level relationships of each subfamily varied significantly across reconstructions based on different matrices. The inclusion of third-codon positions resulted in increased uncertainty of the phylogenetic positions of *A. mihalyii* and *Goniophyto honshuensis* within Paramacronychiinae. When the third-codon positions were excluded, the genus-level relationship within Paramacronychiinae was consistent with the phylogenomic studies [2] and well-supported. However, the pruning of third-codon positions caused instability of relationship within the Sarcophaginae, with either *Blaesoxipha* or *Boettcheria* being a basal branch, compared with (*Blaesoxipha* + *Boettcheria*) being a sister group to the remaining Sarcophaginae when third-codon positions were included. In contrast, the genus-level relationships of Miltogramminae never reached agreement no matter which dataset was used in this study, and the topologies were generally poorly supported. This may be explained by the higher evolutionary rate of Miltogramminae compared to Paramacronychiinae and Sarcophaginae (Figure 3), and the mitochondrial genome may not be sufficiently phylogenetically informative for resolving older splits in the fast-evolving miltogrammines. In addition, the third-codon positions are generally considered less informative and therefore are often excluded in phylogenetic analyses [58,59,60], but a recent study suggests third-codon positions to be phylogenetically informative and recommends that they be included [21]. In the present study, the substitutions of third-codon positions of Miltogramminae and Paramacronychiinae were saturated, compared with unsaturated third-codon positions of the Sarcophaginae (Appendix A), which is likely the reason for the different performance of third-codon positions in phylogenetic analyses of Paramacronychiinae and Sarcophaginae. Taken together, our analyses indicate that the mitochondrial genome may be a suboptimal choice for resolving older phylogenetic splits of fast-evolving organisms, but the third-codon positions could be informative in phylogenetic inference.

There are several genera with multiple mitogenomes from GenBank: *Oxysarcodexia*, *Peckia*, *Sarcophaga*, and *Wohlfahrtia*. The relationships of species within each of these genera were robust and well-supported in all topologies except for *Sarcophaga*, the largest genus of Sarcophagidae. The relationships within *Sarcophaga* varied depending upon whether the third-codon positions or rRNA genes were used (Figure 5 and Appendix A). Such differences may be a result of fast speciation within *Sarcophaga*, as indicated by the short branches for species of this genus. However, the phylogenetic topology for *Sarcophaga* spp. (Figure 5) largely matches the traditional subgeneric divisions [4].

It is interesting to notice that *S. pauciseta* (represented by mitogenomic data with accession of NC_053729) always grouped with *Blaesoxipha*, far from other *Sarcophaga*. This sequence had been generated as part of a project termed “The complete mitochondrial genome of *Sarcophaga rossica*” from an unpublished study in NCBI, while “*rossica*” is the name of a species of *Blaesoxipha* [4]. In addition, the identification of Sarcophaginae mainly relies on the complicated morphological characters of male terminalia [5], and the general adult morphology of *Blaesoxipha* is very similar to that of *Sarcophaga*. Therefore, it is likely that the specimen generating mitogenomic data for *S. pauciseta* was a misidentified specimen of *Blaesoxipha*, which highlights the importance of building DNA barcode libraries for insects, such as flesh flies, that are difficult to identify based on morphological characteristics.

## 4. Conclusions

The current study documents newly sequenced mitogenomes of five species of flesh flies. The mitogenomes examined here are highly conserved in terms of gene content, gene size, gene order, base composition, and codon use of the protein-coding genes, which is consistent with earlier investigations of sarcophagid mitogenomes. Similar to the mitochondrial genomes of other flesh flies, they have a high A + T bias in their nucleotide composition. Among the protein-coding genes, ATP8 appears to have the highest evolutionary rate, while COI has the lowest. Moreover, ND2 and ND6 are shown to exhibit excellent potential as molecular markers for species identification and the construction of phylogenetic relationships. The phylogenetic analysis strongly supports the monophyly of Sarcophagidae and each subfamily, inferring the relationship (Sarcophaginae, (Miltogramminae, Paramacronychiinae)) at the subfamily level, whereas the genus-level phylogeny remains uncertain from the perspective of mitogenomic data. The third-codon positions of the mitochondrial protein-coding genes contribute to the phylogeny construction of Sarcophagidae, particularly within Sarcophaginae.

## Figures and Tables

**Figure 1 insects-13-00718-f001:**
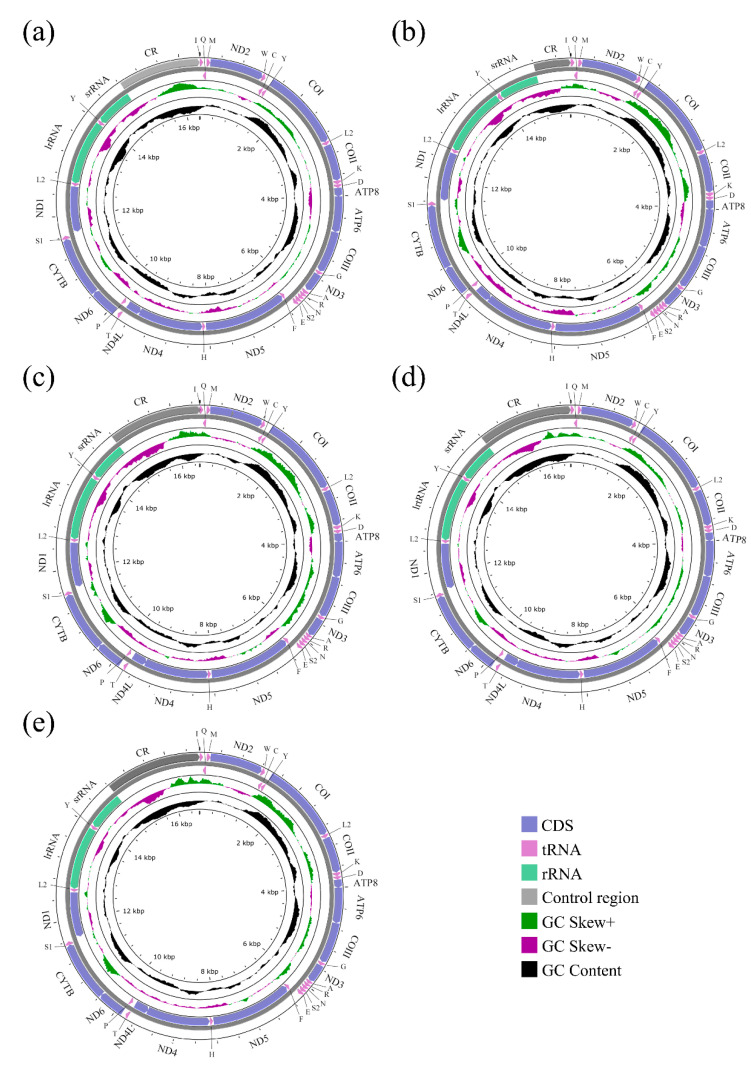
*Agria mihalyii* (**a**), *Blaesoxipha lapidosa* (**b**), *Taxigramma karakulensis* (**c**), *Wohlfahrtia balassogloi* (**d**), and *Wohlfahrtia fedtschenkoi* (**e**). The strands are marked with arrows that indicate the direction of gene transcription. Gene lengths correspond to the length of their nucleotides. The IUPAC-IUB single-letter amino acid codes assign one-letter symbols to tRNA genes. Gene names were represented by their abbreviations: COI–COIII, cytochrome oxidase subunits 1–3; CYTB, cytochrome b; ND1–6 and ND4L, NADH dehydrogenase subunits 1–6 and 4L; ATP6 and ATP8, ATP synthetase subunits 6 and 8; lrRNA and srRNA, large and small ribosomal RNA subunits; CR, control region.

**Figure 2 insects-13-00718-f002:**
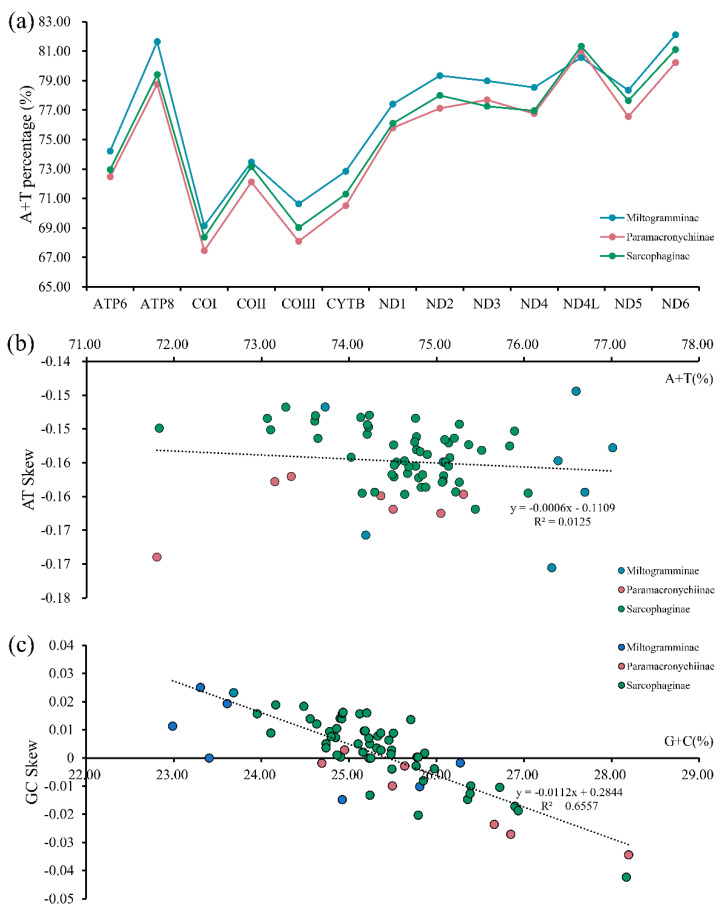
Nucleotide composition analysis of mitochondrial genomes from three subfamilies of Sarcophagidae: A + T percentage of the 13 protein-coding genes (**a**) and the corrections between A + T% vs. AT skew (**b**) and G + C% vs. GC skew (**c**) in the 13 protein-coding genes.

**Figure 3 insects-13-00718-f003:**
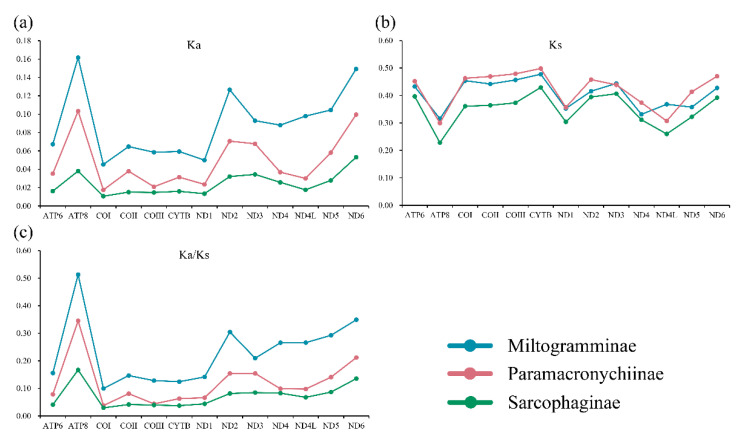
Synonymous (Ka; (**a**)) and nonsynonymous (Ks; (**b**)) substitutional rates and the Ka/Ks ratios (**c**) of the 13 protein-coding genes of the three subfamilies of Sarcophagidae.

**Figure 4 insects-13-00718-f004:**
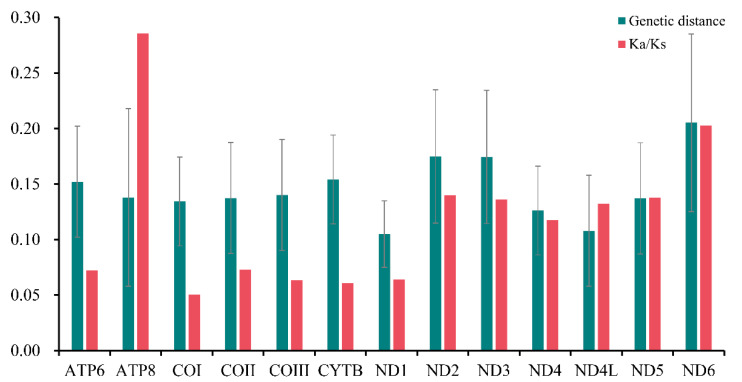
Ratios of nonsynonymous and synonymous substitution (Ka/Ks) and genetic distances of the 13 protein-coding genes of Sarcophagidae. Error bars refer to the standard deviation from all the combined data of all Sarcophagidae.

**Figure 5 insects-13-00718-f005:**
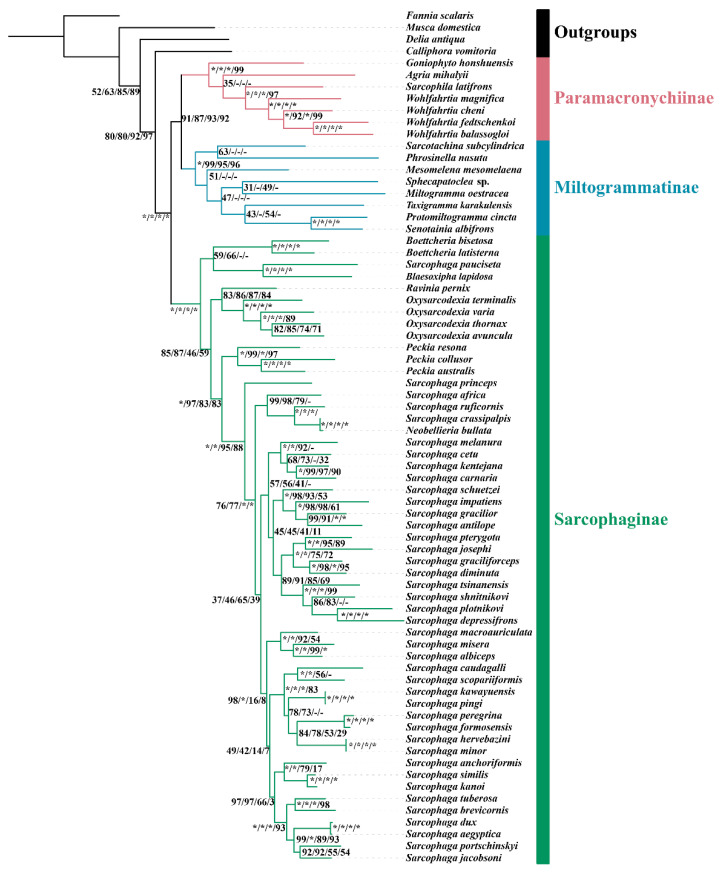
Maximum likelihood phylogenetic tree for Sarcophagidae based on concatenation of 13 mitochondrial protein-coding genes and 2 rRNA genes. Each node is provided with the support values of Maximum Likelihood bootstrap values (BS) for tree inference using matrix PCGsrRNA/PCGs/PCGs12rRNA/PCGs12. Asterisks (*) represent full support, while hyphens (-) indicate that the branch is not recovered.

**Table 1 insects-13-00718-t001:** Taxonomic information and GenBank accession number of mitochondrial genomes used in the study.

Family	Subfamily	Genus	Subgenus	Species	Accession No.
Sarcophagidae	Miltogramminae	*Mesomelena*		*mesomelaena*	KY003227
		*Miltogramma*		*oestracea*	MW556020
		*Phrosinella*		*nasuta*	MW546340
		*Protomiltogramma*		*cincta*	NC_063942
		*Sarcotachina*		*subcylindrica*	MW556014
		*Senotainia*		*albifrons*	MW556016
		*Sphecapatoclea*		sp.	MW556015
		*Taxigramma*		** *karakulensis ** **	ON375459
	Paramacronychiinae	*Agria*		** *mihalyii ** **	ON360967
		*Goniophyto*		*honshuensis*	MW556017
		*Sarcophila*		*latifrons*	MW556018
		*Wohlfahrtia*		** *balassogloi ** **	ON411642
		*Wohlfahrtia*		*cheni*	MW556019
		*Wohlfahrtia*		** *fedtschenkoi ** **	ON411643
		*Wohlfahrtia*		*magnifica*	KU578263
	Sarcophaginae	*Blaesoxipha*		** *lapidosa ** **	OM640654
		*Boettcheria*		*bisetosa*	KT272844
				*latisterna*	KT272848
		*Sarcophaga*	*Bellieriomima*	*diminuta*	MT017719
			*Bellieriomima*	*graciliforceps*	MT017712
			*Bellieriomima*	*josephi*	MT017711
			*Bellieriomima*	*pterygota*	MK820722
			*Bercaea*	*africa*	KM881633
			*Boettcherisca*	*formosensis*	MF688648
			*Boettcherisca*	*peregrina*	KF921296
			*Helicophagella*	*melanura*	KP091687
			*Heteronychia*	*depressifrons*	MT017709
			*Heteronychia*	*pauciseta*	NC_053729
			*Heteronychia*	*plotnikovi*	MT017720
			*Heteronychia*	*shnitnikovi*	MT017730
			*Heteronychia*	*tsinanensis*	NC_057591
			*Kozlovea*	*cetu*	MT017725
			*Kramerea*	*schuetzei*	MT017726
			*Liopygia*	*crassipalpis*	KC005711
			*Liopygia*	*ruficornis*	MH937749
			*Liosarcophaga*	*aegyptica*	MT017732
			*Liosarcophaga*	*brevicornis*	NC_047404
			*Liosarcophaga*	*dux*	MH879759
			*Liosarcophaga*	*jacobsoni*	MT017723
			*Liosarcophaga*	*kanoi*	MT476487
			*Liosarcophaga*	*portschinskyi*	KM287570
			*Liosarcophaga*	*tuberosa*	MK820723
			*Neobellieria*	*bullata*	KT272859
			*Pandelleisca*	*kawayuensis*	MT017713
			*Pandelleisca*	*pingi*	MT017728
			*Pandelleisca*	*scopariiformis*	MT476486
			*Pandelleisca*	*similis*	KM287431
			*Parasarcophaga*	*albiceps*	KT444443
			*Parasarcophaga*	*macroauriculata*	MT017718
			*Parasarcophaga*	*misera*	MF133500
			*Phallocheira*	*minor*	MT017727
			*Pseudothyrsocnema*	*caudagalli*	MK820721
			*Robineauella*	*anchoriformis*	MT017716
			*Sarcophaga*	*carnaria*	MT017710
			*Sarcorohdendorfia*	*antilope*	MH540748
			*Sarcorohdendorfia*	*gracilior*	MW531675
			*Sarcorohdendorfia*	*impatiens*	JN859549
			*Seniorwhitea*	*princeps*	MH981944
			*Sinonipponia*	*hervebazini*	MT017708
			*Thyrsocnema*	*kentejana*	MT017714
		*Ravinia*		*pernix*	KM676414
		*Oxysarcodexia*		*avuncula*	MH879754
		*Oxysarcodexia*		*terminalis*	MH879757
		*Oxysarcodexia*		*thornax*	MH879765
		*Oxysarcodexia*		*varia*	MH879764
		*Peckia*		*australis*	MH879762
		*Peckia*		*collusor*	MH879763
		*Peckia*		*resona*	MH879761
Calliphoridae		*Calliphora*		*vomitoria*	KT444440
Anthomyiidae		*Delia*		*antiqua*	NC_028226
Fanniidae		*Fannia*		*scalaris*	MT017706
Muscidae		*Musca*		*domestica*	KM200723

* The sequences for species with names in bold were generated in this study.

## Data Availability

The five newly sequenced mitogenomes were submitted to the GenBank database under the accession numbers of *Agria mihalyii* (ON360967), *Blaesoxipha lapidosa* (OM640654), *Taxigramma karakulensis* (ON375459), *Wohlfahrtia balassogloi* (ON411642), and *Wohlfahrtia_fedtschenkoi* (ON411643). The associated raw data produced for this study can be found in the NCBI Sequence Read Archive (SRA) under accession numbers PRJNA684426 and PRJNA844010.

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
