# Peer review of "Comparative Mitogenomics of Flesh Flies: Implications for Phylogeny"

_insects, 2022, doi:10.3390/insects13080718_

Round 1

Reviewer 1 Report

The manuscript by Shang et al. “Comparative mitogenomics of flesh flies: implications for phylogeny” reports the sequence of the mitogenomes of five species of flesh flies and performs a phylogenetic study with all the available mitogenomes of this group of flies. The subject may be of interest for readers, but the paper has some shortcomings.

I think that the material and methods section should provide more details (see examples below).

Some calculations are not well thought out. For example, the sliding windows of Pi reported on figure 4b don’t make any sense because the gene sequences are concatenated in an order (alphabetically) and sense that is different from the actual one in the mitogenome. Some windows will take part of the sequence from one gene and part of the sequence from the next, when in fact they are not contiguous.

I also have doubts about obtaining a phylogenetic tree based on PCGs12. Normally, the synonymous variation is chosen to reconstruct phylogenies. However, position 2 is always a nonsynonymous position, and position 1 is also nonsynonymous in most cases. Thus, positions 1 and 2 should be under a higher evolutionary constraint that should make it difficult to correctly separate closely related species.

Comments to the authors:

The Materials and Methods section needs more details:

L. 87-88. Was the genomic DNA from the five species pooled together? If so, how were the sequences later separated? Are the differences among the five species enough to obtain five independent contigs? Was the entire genome (nuclear and mitochondrial) sequenced? What about the x coverage?

L103-105. More details on how the AT skew and the GC skew were calculated should be provided. I think that they were calculated only on PCGs, but in their full length or only in the fourfold degenerate third codon position sites?

In the firsts sections of Results and Discussion, I have found it difficult to understand when authors refer only to the five newly sequenced mitogenomes and when they refer to the entire sample of analyzed mitogenomes. For example:

L.139-140. “The variation in length is mainly due to the different size of the control region, as per earlier studies [38].” This sentence is confusing as it seems to indicate that the differences between these five mitogenomes were already known. It should be better to say: “The variation in length is mainly due to the different size of the control region, as previously observed between other Sarcophagidae species [38]”.

L.140-150, when I read these lines the first time, I thought they referred to the five sequenced mitogenomes, but after reading the following paragraphs I’m not sure if they refer to the 68 species.

L. 159-165. This paragraph begins with data from the five new sequenced mitogenomes, but it ends with a global view of the three subfamilies. I think that the transition from one idea to the other is not clear, and I had to read several times to understand it.

L.221-223, “Therefore, the length of the mitochondrial genome of sarcophagid species mainly depends on the size of the control region, which is consistent with other insects (Table S2).” This sentence should be rewritten. For context, “therefore” here should refer only to the five newly sequenced species, not the sarcophagid species in general. In addition, Table S2 only shows data for these five species, not for other insects as it seems to reference.

L.234-237 “The COI had the lowest Ka/Ks value (ω 234 = 0.029) in Paramacronychiinae, which could be associated with strong purifying selection. ATP8 exhibited the highest rate (ω = 0.513) in Miltogramminae, which is indicative of relaxed purifying selection.

Looking at Figure 3 the lowest Ka/Ks value seems to be for COI but for Sarcophaginae. On the other hand, I think the important thing is which gene has the lowest and highest values. COI is the gene with the lowest value in all three families and ATP8 is the gene with the highest values in all three families.

L.258-259 References 55, 56 and 57 are not necessary. The relevance of the Pi statistics is well-known.

L.262-264. “The near-consistent results of the pairwise genetic distance and the nucleotide diversity analysis further suggest that ND6, ND2 and ND3 evolved at faster evolutionary rates or with higher variability.”

Results of these two parameters should be coincident since are based in the same. The genetic distance that authors have obtained with MEGA and Kimura 2-parameters is a correction of Pi (nucleotide diversity) for multiple hits considering differently transitions and transversions.

Minor comments and typos.

L.204 Change “contents” to “content”

L.205 Change “tRNA gene to “tRNA genes”

L.206 Change “protein-coding gene to 2protein-coding genes”

Figure 1. The definition of the figure is not good and the arrows indicated by the foot of the figure are not appreciated.

Figure S1. It is difficult to identify the name of the species. Maybe those of the five new sequenced could be written in red or in bold face.

Reviewer 2 Report

This is a manuscript of high quality. The analyses are comprehensive and the documentation itself is exhaustive. Acceptance of publication is recommended. The minor comments I have is that some figure legends need to be more elaborate. For example, what do the errors bars represent in Fig4a?

L-272 "slowest evolving" should be "most slowly evolving"

Round 2

Reviewer 1 Report

The new version of the paper “Comparative mitogenomics of flesh flies: implications for phylogeny” has been much improved. In addition to responding to all the technical points requested, the writing is much clearer, and the reading is easier to follow.

However, I would just like to highlight the following additional minor aspects:

L. 141-143 “The entire mitogenomes of A. mihalyii, B. lapidosa, T. karakulensis, W. balassogloi, and W. fedtschenkoi, which were 16375 bp, 15454 bp, 16543 bp, 16589 bp, and 16643 bp in length, respectively (Figure 1).

In this new version, there is the addition of a “which” that should be deleted. Right now, the sentence seems incomplete.

L. 221-223 “Extrapolated from the information of these five species, the length of the mitochondrial genome of sarcophagid species mainly depends on the size of the control region, which is consistent with other insects (Table S2).

Referring to Table S2 at the end of the sentence (after “which is consistent with other insects”), seems indicate that this Table give information from other insects. It should be better changing the order:

“Extrapolated from the information of these five species (Table S2), the length of the mitochondrial genome of sarcophagid species seems to depend mainly on the size of the control region, which is consistent with other insects.”

L. 228-230 “All three subfamilies have similar synonymous substitution rates (Ks), while the non-synonymous substitution rates (Ka) of Miltogramminae were significantly higher than those of Paramacronychiinae and Sarcophaginae (Figure 3).” Are the values of Ka of Miltogramminae really significantly higher? The scales of the figures 3a and 3b are very different. I would change “significantly” to “in all cases”, for example.

L.238-246. This paragraph should be deleted, since there is a new version on lines 250-260. 

On the other hand, there are two things I don’t quite understand on Supplementary Material: 

1.     What is the order of the species in the figure S1? It might seem like that they are in alphabetical order within each subfamily (as in Table 1), but they are not. For example, there is the following series: … Senotainia albifrons/ Agria mihalyii / Blaesoxipha lapidosa / Boettcheria latisterna / Goniophyto honshuensis /Mesomelena mesomelaena ….These species are from subfamilies …M/P /S /S / P /M…

2.     Figure S2. I don’t understand the data for Leu1-Leu2 and Ser1-Ser2. If each one of these aa has 4 possible codons, why the high of they the bars for version 1 and version 2 of these aa have less that 4 and more than 4, respectively?
